# Identification of Novel *tet*(X3) Variants Resistant To Tigecycline in *Acinetobacter* Species

Yumeng Cheng,[a] Yakun Li,[b] Runhao Yu,[c] Mingxiang Ma,[a] Meng Yang,[a] Hongbin Si[a]

[a]College of Animal Science and Technology, State Key Laboratory for Conservation and Utilization of Subtropical Agro-Bioresources, Guangxi University, Nanning, China
[b]College of Life Science and Technology, State Key Laboratory for Conservation and Utilization of Subtropical Agro-Bioresources, Guangxi University, Nanning, China
[c]College of Veterinary Medicine, Henan Agricultural University, Zhengzhou, China

**ABSTRACT**   The emergence of the *tet*(X) gene is a severe challenge to global public health security, as clinical tigecycline resistance shows a rapidly rising trend. In this research, we identified two tigecycline-resistant *Acinetobacter* sp. strains containing seven novel *tet*(X3) variants recovered from fecal samples from Chinese farms. The seven Tet(X3) variants showed 15.4% to 99.7% amino acid identity with Tet(X3). By expressing *tet*(X3.7) and *tet*(X3.9), the tigecycline MIC values for *Escherichia coli* JM109 increased 64-fold (from 0.13 to 8 mg/L). However, the other *tet*(X3) variants did not have a significant change in the MIC of tigecycline. We found that the 26th amino acid site of Tet(X3.7) changed from proline to serine, and the 25th amino acid site of Tet(X3.9) changed from glycine to alanine, which reduced the MIC of tigecycline by 2-fold [the MIC of *tet*(X3) to tigecycline was 16 mg/L] but did not affect its expression to tigecycline. The *tet*(X3) variants surrounded by mobile genetic elements appeared in the structure of gene clusters with tandem repeat sequences and were adjacent to the site-specific recombinase-encoding gene *xerD*. Therefore, there is a risk of horizontal transfer of resistant genes. Our study reports seven novel *tet*(X3) variants; the continuing emergence of tigecycline variants makes continuous monitoring of resistance to tigecycline even more critical.

**IMPORTANCE**   Although it is illegal to use tigecycline and carbapenems to treat bacterial infections in animals, we can still isolate bacteria containing both mobile resistance genes from animals, and *tet*(X) is currently an essential factor in degrading tigecycline. Here, we characterized two multidrug-resistant *Acinetobacter* sp. strains that contained vital resistance genes, such as *sul2*, a *bla*$_{OXA-164}$-like gene, *floR*, *tetM*, and multiple novel *tet*(X3) variants with different tandem structures. It is of paramount significance that their mechanism may transfer to other Gram-negative pathogens, even if their tandem structures have no cumulative effect on tigecycline resistance.

**KEYWORDS**   tigecycline resistance, *tet*(X), *tet*(X3), *Acinetobacter variabilis*, *Acinetobacter schindleri*

Address correspondence to Hongbin Si, shb2009@gxu.edu.cn.

The authors declare no conflict of interest.

Carbapenem-resistant *Enterobacteriaceae* (CRE) and *Acinetobacter* spp. (CRA) are currently one of the greatest threats to global public safety. Tigecycline and colistin are the two last resorts for defending against CRE and CRA infections (1). At the same time, with the widespread use of these antibiotics, strains carrying tigecycline- and colistin-resistant genes have emerged in clinical settings (2).

Tigecycline is an extended-spectrum glycylcycline that can avoid tetracycline resistance mediated by an active efflux pump (AcrAB-TolC, MexXY-OprM, and OqxAB) or ribosomal protection (*rpsJ*, *plsC*, and *trm* of mutations) in *Enterobacteriaceae* members (3–11). It has a broad spectrum of activity in multidrug-resistant/extensively drug-resistant (MDR/XDR) Gram-positive and Gram-negative bacteria, except for *Proteus* spp. and *Pseudomonas aeruginosa* (12). Tigecycline is considered one of the last-resort treatments for severe infections

**TABLE 1** MICs of the parental strain, transconjugants, and transformants[a]

| Strain | MIC (mg/L) | | | | | | | | |
|---|---|---|---|---|---|---|---|---|---|
| | CIP | IMP | MEM | FFC | STR | ERY | FOS | TC | TGC |
| BDT2044 | 16 | 64 | 32 | 256 | 32 | 64 | 512 | 128 | 32 |
| BDT2091 | 512 | 64 | 32 | 128 | 64 | 128 | 512 | 128 | 16 |
| ADP1 | <1 | 0.25 | <1 | <1 | <1 | <1 | <1 | 0.5 | 0.25 |
| JAT2044 | <1 | 0.5 | <1 | <1 | <1 | <1 | <1 | 4 | 8 |
| JAT2091 | <1 | 0.5 | <1 | <1 | <1 | <1 | <1 | 4 | 4 |
| JM109 | | 0.25 | <1 | | | | | 1 | 0.13 |
| JM109+pBAD24[b] | | 0.25 | <1 | | | | | 1 | 0.13 |
| JM109+pBAD24-*tet*(X3) | | | | | | | | 16 | 16 |
| JM109+pBAD24-*tet*(X3.3) | | | | | | | | 1 | 0.13 |
| JM109+pBAD24-*tet*(X3.4) | | | | | | | | 1 | 0.13 |
| JM109+pBAD24-*tet*(X3.5) | | | | | | | | 1 | 0.13 |
| JM109+pBAD24-*tet*(X3.6) | | | | | | | | 1 | 0.13 |
| JM109+pBAD24-*tet*(X3.7) | | | | | | | | 8 | 8 |
| JM109+pBAD24-*tet*(X3.8) | | | | | | | | 1 | 0.13 |
| JM109+pBAD24-*tet*(X3.9) | | | | | | | | 8 | 8 |

[a]CIP, ciprofloxacin; IMP, imipenem; MEM, meropenem; FFC, florfenicol; STR, streptomycin; ERY, erythromycin; FOS, fosfomycin; TC, tetracycline; TGC, tigecycline. Empty cells indicate that the MIC was not determined.
[b]Plasmid pBAD24 is an empty vector.

caused by MDR, XDR, and pandrug-resistant (PDR) strains of Gram-negative pathogens (13). However, with the widespread clinical application of tetracycline, the emergence and spread of mobile and high levels of the tigecycline resistance genes *tet* (X3), *tet* (X4), and other *tet*(X) variants are altering this status (1, 14–22). Variants of the tetracycline efflux pump genes *tet*(A), *tet*(M), and *tet*(L) have been reported to cause tigecycline resistance (23–26). Tet(X), the flagship tetracycline-inactivating enzyme (27), a flavin-dependent monooxygenase modified by 388 amino acids encoding tetracycline, is a novel resistance gene that directly inactivates tetracycline and can only be activated in the presence of FAD, NADPH, $Mg^{2+}$, and $O_2$ (28–30). It was first discovered in the R plasmid of the obligate anaerobe human symbiotic bacterium *Bacteroides fragilis* in the 1980s (31, 32). The emergence and rapid spread of antimicrobial resistance in *Enterobacteriaceae* members poses a severe threat to human and animal health (33, 34).

In this study, we screened and identified *tet*(X)-related resistance genes in strains recovered from fecal samples on a pig farm and a chicken farm in Guangxi Zhuang Autonomous Region, China. *Acinetobacter variabilis* BDT2044 and *Acinetobacter schindleri* BDT2091 were successfully isolated, with multiple *tet*(X) variant genes exhibiting high resistance to tigecycline, and validated separately.

## RESULTS

**Characterization of tigecycline-resistant isolates.** The tigecycline-resistant isolates *Acinetobacter variabilis* BDT2044 and *Acinetobacter schindleri* BDT2091 were recovered from fecal samples from a pig farm and a chicken farm, respectively, in Guangxi Zhuang Autonomous Region, China. By the broth microdilution method, they were resistant to ciprofloxacin, imipenem, meropenem, florfenicol, streptomycin, erythromycin, fosfomycin, tetracycline, and tigecycline (Table 1). The two isolates are thus MDR bacteria.

**WGS and genetic background analysis of BDT2044 and BDT2091.** Whole-genome sequencing (WGS) of BDT2044 showed that it contained one 3,345,806-bp chromosome and seven plasmids with sizes of 10,548 bp (pBDT2044-1), 17,439 bp (pBDT2044-2), 18,754 bp (pBDT2044-3), 20,030 bp (pBDT2044-4), 36,028 bp (pBDT2044-5), 46,646 bp (pBDT2044-6), and 55,901 bp (pBDT2044-7). The conjugative plasmid pBDT2044-7 contained the resistance genes *tetM*, *floR*, *bla*$_{OXA-164}$, *aac*(3)-*IId*, and *aph*(3')-*Ia* (see Fig. S1 in the supplemental material), and these resistance genes were further verified by PCR identification and sequencing. The chromosome contained a genomic island of 83,342 bp containing 13 consecutive *tet*(X3) tigecycline resistance genes that were adjacent to two copies of IS*Vsa3* elements and a ΔIS*CR2* element, in which there were four novel *tet*(X3) variants

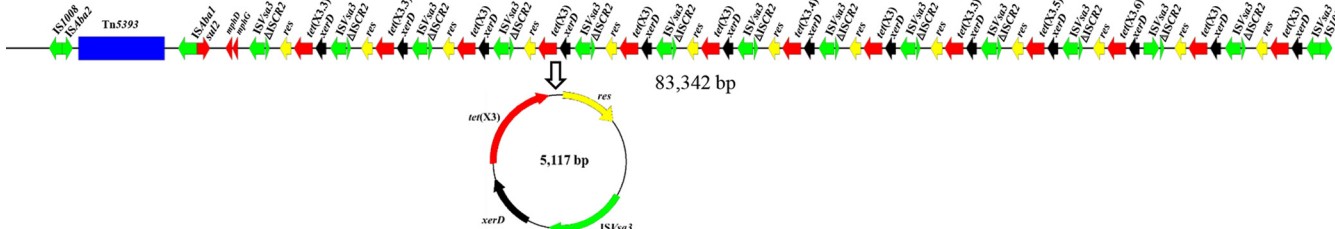

**FIG 1** Genomic island structure of the BDT2044 chromosome and minicircle plasmid of JAT2044. Positions and transcriptional directions of the predicted open reading frames (ORFs) are denoted with arrows. Genes associated with antimicrobial resistance, mobile elements, resolvase, recombinase, and the transposon are highlighted in red, green, yellow, black, and blue, respectively. The repeat sequence was the IS*Vsa3*-ΔIS*CR2*-*res*-*tet*(X3)-*xerD* gene cassette.

(Fig. 1). We compared the four novel *tet*(X3) variants in this article with those found at GenBank, and the genetic background sequences of our isolates were similar to those found under accession numbers CP060813, CP044446, CP084302, and MK134375 (Fig. 2).

Further analysis showed that the *tet*(X3) variants were closely related to the genetic elements IS*Vsa3* and IS*CR2* on the genomic island. Intriguingly, on the same genomic island, the chromosome harbored 13 tandem repeated gene cassettes, IS*Vsa3*-ΔIS*CR2*-*res*-*tet*(X3)-*xerD*. To determine whether this region could also be mobile, a potentially mobile 5,117-bp circular intermediate consisting of the *tet*(X3) variant carried in the central region was examined. Only one copy of the IS*Vsa3* element [IS*Vsa3*-*xerD*-*tet*(X3)-*res*-ΔIS*CR2*) was created by inverse PCR, suggesting that the *tet*(X3) variants harboring genetic elements were highly active. They may further transfer to other plasmids or chromosomes.

WGS of BDT2091 showed that it contained one 3,135,005-bp chromosome and six plasmids with sizes of 11,814 bp (pBDT2091-1), 17,364 bp (pBDT2091-2), 18,530 bp

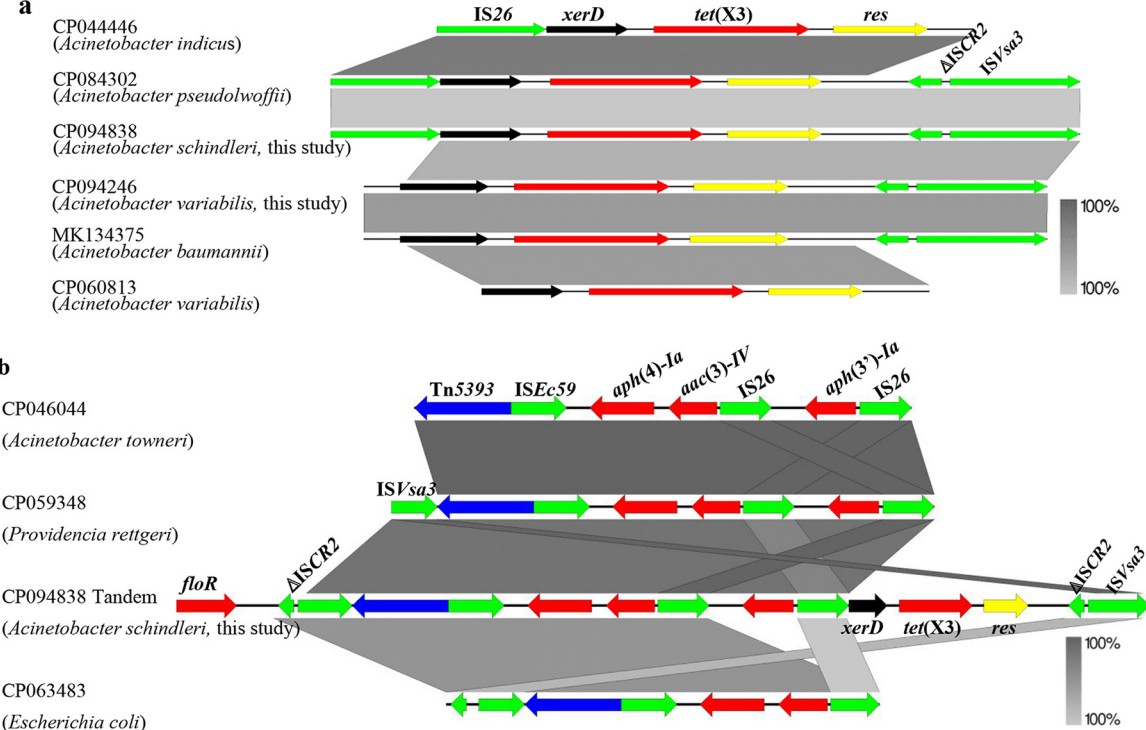

**FIG 2** Genetic structure in this study and comparison with similar regions in sequences deposited at GenBank. (a) Genetic structure of *tet*(X3) in this research compared with similar regions in sequences deposited under GenBank accession numbers CP060813, CP044446, CP084302, and MK134375. (b) Genetic structure of plasmid-borne tandem repeats in pBDT2091-4 was compared with similar regions in CP046044, CP059348, and CP063483. Regions of homology up to 100% are indicated by gray shading. Positions and transcriptional directions of the predicted ORFs are denoted by arrows. Genes associated with antimicrobial resistance, mobile elements, resolvase, recombinase, and transposon are highlighted in red, green, yellow, black, and blue, respectively.

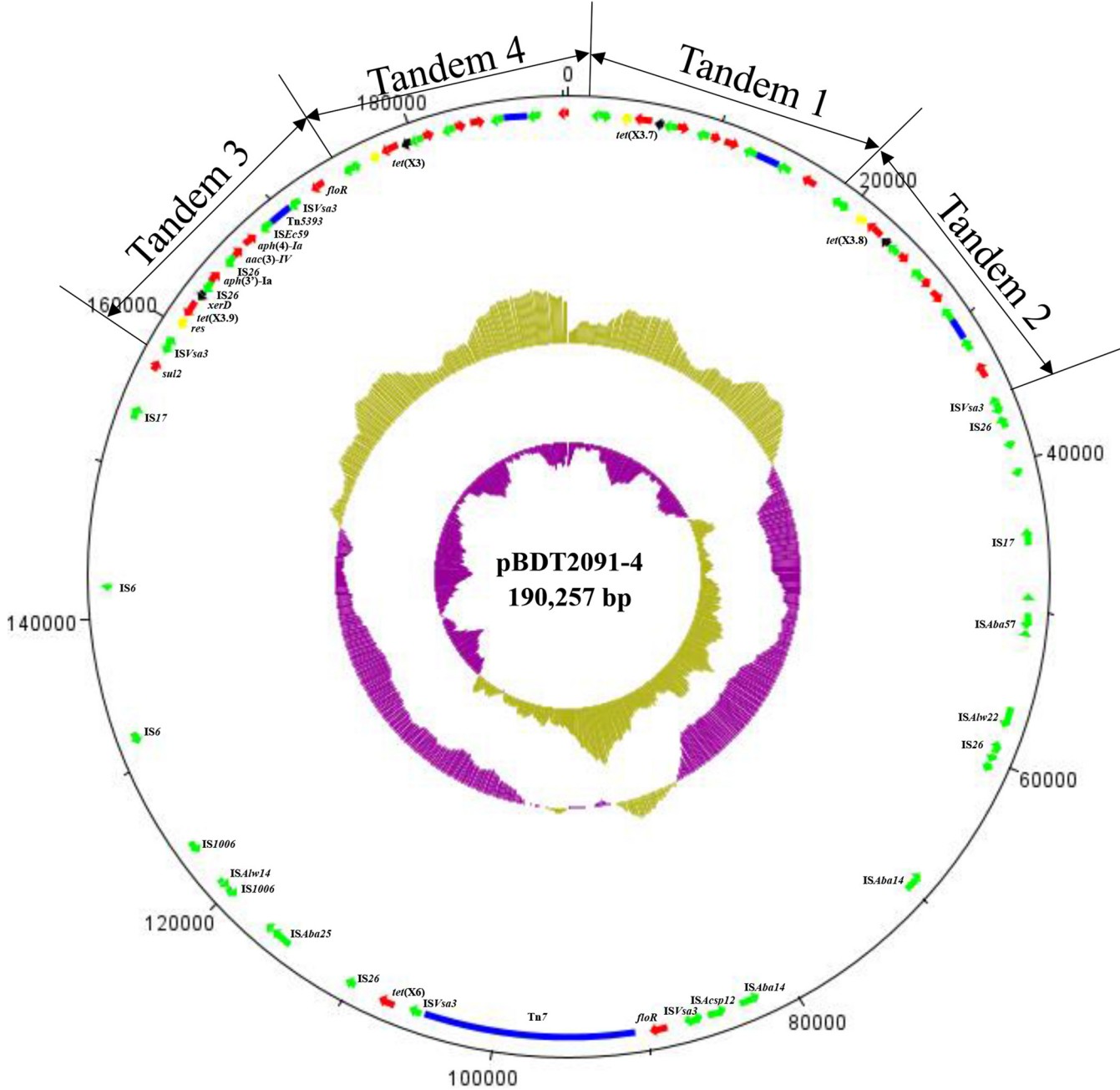

**FIG 3** Structure of plasmid pBDT2091-4. (Inner to outer circles) GC skew and GC content are indicated. Positions and transcriptional directions of the predicted ORFs are denoted with arrows. Genes associated with antimicrobial resistance, mobile elements, resolvase, recombinase, and transposons are highlighted in red, green, yellow, black, and blue, respectively. Four repeated sequences are labeled Tandem 1 to 4. The repeat sequence is the structure IS*Vsa3*-ΔIS*CR2-res-tet*(X3)-*xerD*-IS*26-aph*(3')-*Ia*-IS*26-aac*(3)-*IV-aph*(4)-*Ia*-IS*Ec59*-Tn*5393*-IS*Vsa3-floR*.

(pBDT2091-3), 190,257 bp (pBDT2091-4), 19,542 bp (pBDT2091-5), and 50,645 bp (pBDT2091-6). Plasmid pBDT2091-6 was a conjugative plasmid containing the *bla*_{OXA-23} gene (Fig. S2). The results revealed that the tigecycline resistance genes *tet*(X3), *tet*(X6), and three novel *tet*(X3) variants were located on plasmid pBDT2091-4 (Fig. 3). The tigecycline-resistant *tet*(X3) variants were located near transposon Tn*5393* and were adjacent to the genetic elements IS*Vsa3* and IS*26* (Fig. 3). Three of the *tet*(X3) variants in this research were compared with the GenBank database, and the genetic background sequences were found to be similar to those of CP060813, CP044446, CP084302, and MK134375 (Fig. 2a). Intriguingly, plasmid pBDT2091-4 harbored 4 tandem repeated genetic structures, IS*Vsa3*-ΔIS*CR2-res-tet*(X3)-*xerD*-IS*26-aph*(3')-*Ia*-IS*26-aac*(3)-*IV-aph*(4)-

*la*-IS*Ec59*-Tn*5393*-IS*Vsa3*-*floR*. Comparison with sequences at GenBank indicated that the tandem repeated sequences were similar to those of CP046044, CP059348, and CP063483 (Fig. 2b).

These seven novel *tet*(X3) variants were designated *tet*(X3.3), *tet*(X3.4), *tet*(X3.5), *tet*(X3.6), *tet*(X3.7), *tet*(X3.8), and *tet*(X3.9) and had lengths of 1,166 bp, 1,164 bp, 1,165 bp, 1,165 bp, 1,167 bp, 1,165 bp, and 1,167 bp, respectively. The novel Tet(X3) variants had 58.78%, 66.13%, 22.71%, 23.28%, 99.7%, 15.4%, and 99.7% amino acid identity compared with previously reported Tet(X3), respectively (Fig. 4). Phylogenetic analysis indicated that four of the novel Tet(X3) variants formed a clade distinct from the reported Tet(X) variants (Fig. 5).

**Results of conjugation transfer.** Further conjugation assays showed that the transconjugants were not successfully transferred to the recipients *Acinetobacter baumannii* ATCC 19606, *Escherichia coli* EC600, or *E. coli* J53 after three repeats. However, the two transconjugants, designated JAT2044 and JAT2091, did conjugate when transferred with *Acinetobacter baylyi* ADP1 at frequencies of $(5.3 \pm 0.4) \times 10^{-6}$ and $(1.2 \pm 0.8) \times 10^{-7}$ cells per recipient cell, respectively. The MICs of tigecycline increased at least 16- to 32-fold compared with the recipient strain ADP1 (Table 1).

Analysis of the tigecycline-resistant transconjugants revealed that the 5,117-bp circular intermediate [*res*-ΔIS*CR2*-IS*Vsa3*-*xerD*-*tet*(X3)] formed by BDT2044 shedding and entering ADP1 formed a minicircle plasmid (Fig. 1) that mediated the resistance of the transconjugant JAT2044 to tigecycline, which confirmed that the transfer of the *tet*(X3) occurred through the element IS*Vsa3*. Meanwhile, comparative analysis revealed that a region of plasmid pBDT2091-4 reformed as a 27,995-bp plasmid named pJAT2091 in transconjugant JAT2091 and mediated the resistance of transconjugant JAT2091 to tigecycline (Fig. 6). Plasmid pJAT2091 was formed by transferring some resistance regions of pBDT2091-4 with the help of the conjugative plasmid pBDT2091-6, indicating that pBDT2091-4 was nonconjugative but that in its structure, some resistance genes could be mobilized. The tigecycline resistance region was transferred into the recipient strain ADP1 by mobilizing the conjugative plasmid pBDT2091-6.

**Functional confirmation of genes.** Sequence alignment found that these novel *tet*(X3) variants were 1 to 3 bases less than *tet*(X3). There was a mutation of the 76th base from C to T in *tet*(X3.7). There was also a mutation of the 74th base from G to C in *tet*(X3.9).

To determine whether these novel *tet*(X3) variants mediated tigecycline resistance, we cloned the promoter region and gene into pBAD24 to construct pBAD24-*tet*(X3) and pBAD24-*tet*(X3) variants and transferred them into tigecycline-susceptible *E. coli* JM109. However, only JM109+pBAD24-*tet*(X3.7), JM109+pBAD24-*tet*(X3.9), and JM109+pBAD24-*tet*(X3) showed increased resistance—a 64- to 128-fold increase in the MIC of tigecycline and an 8- to 16-fold increase in the MIC of tetracycline—compared to *E. coli* JM109 carrying pBAD24, while the remaining *tet*(X3) variants did not have an altered MIC of tigecycline (Table 1).

## DISCUSSION

Tigecycline has not been used in veterinary clinics. However, tetracyclines were widely used in treating infections in animals in China, which may provide selection pressure for the emergence of tigecycline resistance (35). The tigecycline-resistant *tet*(X) variants have already been extended from *tet*(X3) to *tet*(X47). The *mcr* colistin resistance genes have already been extended to *mcr-10* and have been disseminated globally (36–38). Furthermore, most of these resistance genes can be disseminated through horizontal transfer, significantly damaging global public health security again. Monitoring and controlling the dissemination of such resistance genes is one efficient strategy to combat antimicrobial resistance in both animal and human clinical strains in the context of One Health (considering the human, animal, and environmental sectors) (39, 40). Identifying the key factors of these resistance genes is crucial for fulfilling the strategy and may help to improve the current control measures.

In this study, seven novel variants of *tet*(X3) resistance genes were identified on the chromosome or plasmids of the tigecycline-resistant strains *A. variabilis* BDT2044 and

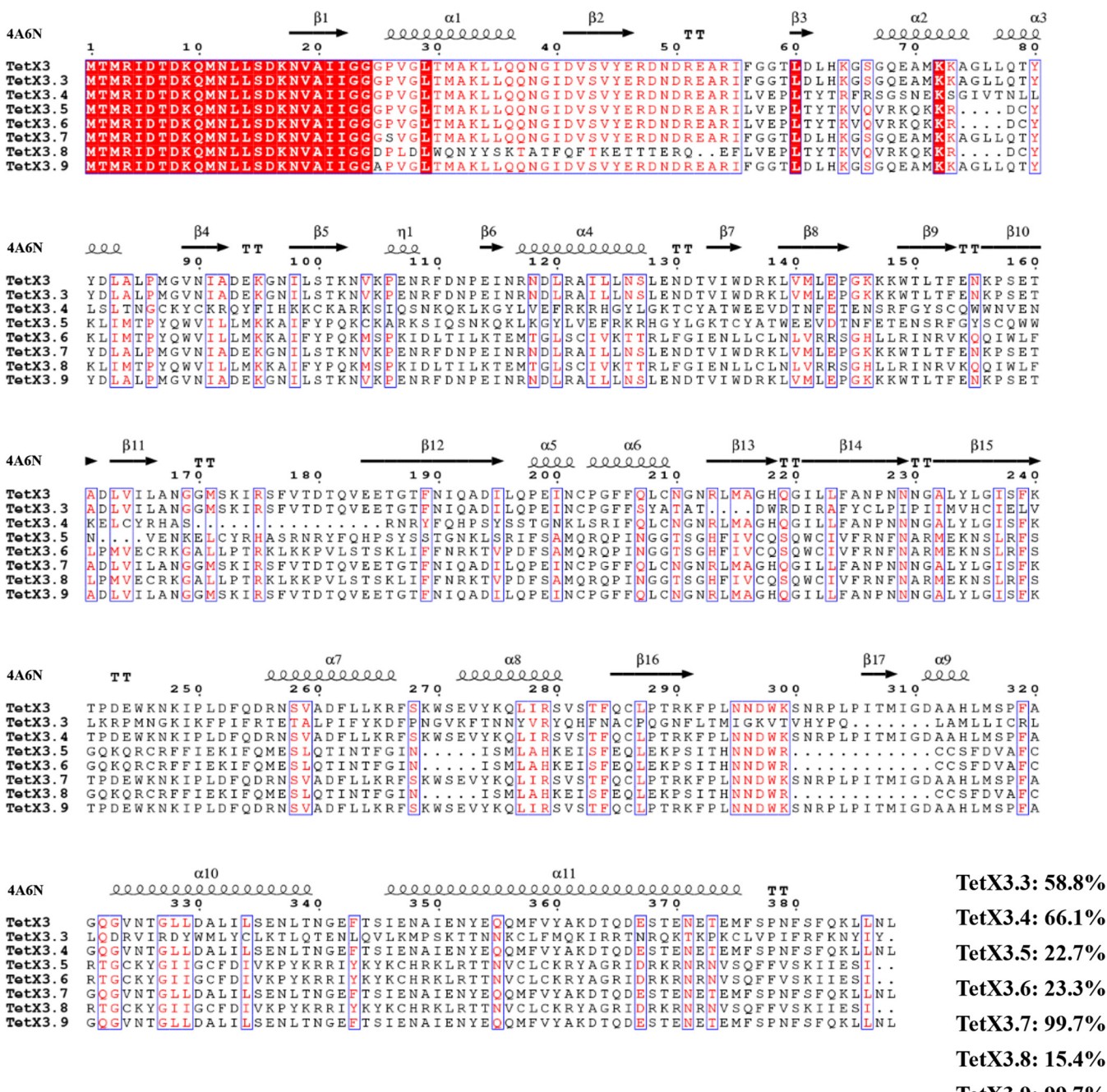

FIG 4 BLAST alignments of amino acid sequences of eight Tet(X3) variants, constructed using ClustalW version 2.1 and ESPript version 3.0 (https://www.genome.jp/tools-bin/clustalw and http://espript.ibcp.fr/ESPript/cgi-bin/ESPript.cgi). The percentages represent the amino acid identity of the novel Tet(X3) variants to the previously reported Tet(X3). The structure found under PDB accession number 4A6N served as the reference for secondary structure depiction.

*A. schindleri* BDT2091, recovered from fecal samples in Guangxi Zhuang Autonomous Region, China. Conjugation assays showed that the tigecycline resistance gene could be successfully transferred from *A. variabilis* BDT2044 and *A. schindleri* BDT2091 into *Acinetobacter baylyi* ADP1, so that the MIC of tigecycline was increased, which is similar to a previous report (41). At the same time, only ADP1 was successfully transferred by conjugation as a recipient strain, indicating that the plasmids can only replicate in *A. baylyi* and that the *tet*(X3) variants can be successfully transferred into *A. baylyi* ADP1, which is also of clinical significance. Conjugation assays using *E. coli* as the recipient strain failed. The species variation of *E. coli* and *Acinetobacter* spp. might affect

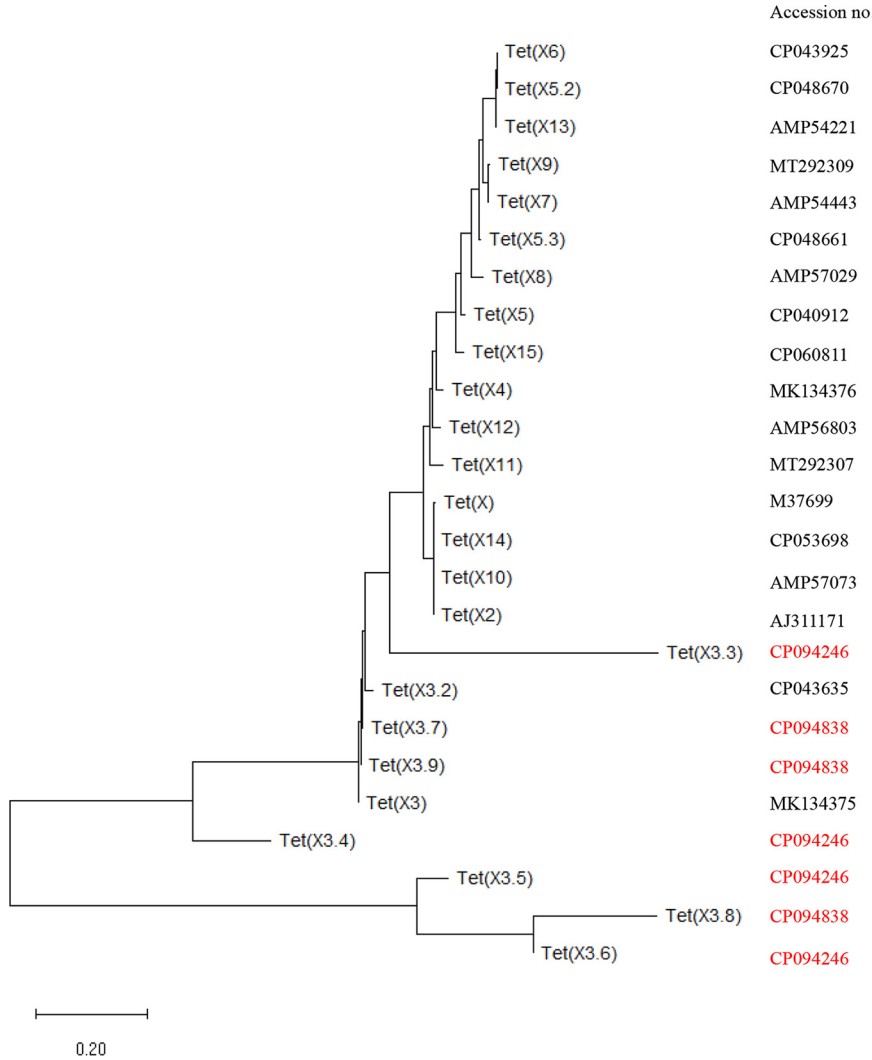

Accession no.

**FIG 5** Phylogenetic analysis of amino acid sequences of different Tet(X) variants. Phylogenetic analysis of the amino acid sequences of all Tet(X) variants was conducted using the neighbor-joining method, using MEGA version 11.0.11 with default parameters and 500 bootstraps. The GenBank accession numbers of the Tet(X) variants are listed [those of the seven novel Tet(X3) variants in this study are shown in red].

horizontal transfer of the plasmids. Using inverse PCR, it was shown that the chromosome of *A. variabilis* BDT2044 formed a 5,117-bp circular intermediate that consisted of the *tet*(X3) variants containing the central region and only one copy of the ISVsa3 element, consistent with a previous study (14). This region was highly mobile, suggesting that these mobile tigecycline resistance genes may be introduced into human

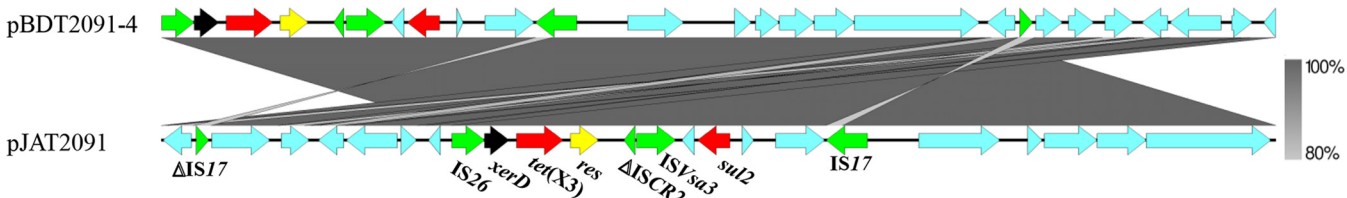

**FIG 6** Genetic environment of *tet*(X3) in plasmid pJAT2091 and comparison with the *tet*(X3)-carrying regions in pBDT2091-4. Regions of homology from 80% to 100% are marked by gray shading. The positions and transcriptional directions of the predicted ORFs are denoted with arrows. Genes associated with antimicrobial resistance, mobile elements, resolvase, recombinase, and other hypothetical proteins are highlighted in red, green, yellow, black, and azure, respectively.

pathogens to create clinically resistant strains, resulting in untreatable infections. The *tet*(X3) variants in pBDT2091-4 were also adjacent to different insertion sequence (IS) genetic elements, which could not recombine with the circular intermediate, according to inverse PCR. However, the tigecycline resistance genes can also be transmitted by horizontal transfer.

Interestingly, the gene experiment showed that *tet*(X3.7) and *tet*(X3.9) in *E. coli* JM109 demonstrated a 64-fold increase in the MIC of tigecycline. In contrast, other *tet*(X3) variants in this research did not contribute to tigecycline resistance. This suggests that mutation of the 74th base from G to C leads to a change of the 25th amino acid from glycine to alanine, and mutation of the 76th base from C to T leads to a change of the 26th amino acid from proline to serine, which still lead to the degradation of tigecycline, maintaining resistance. At the same time, the other amino acid changes may not inactivate tigecycline, so bacteria remain sensitive to tigecycline.

In conclusion, we report the presence of seven novel *tet*(X3) variants, some of which have the potential to be mobile and confer resistance to tigecycline in the parental strain or the *E. coli* host. Whether *tet*(X3) variants are located on the chromosomes or plasmids of isolates, they may spread mediated by mobile elements. Therefore, this research shows that monitoring the constant appearance of *tet*(X3) variants is of great significance for animal health, as it will allow us to develop effective countermeasures against these variants.

## MATERIALS AND METHODS

**Collection of samples and bacterial identification.** In 2020, 179 samples were collected from a pig farm ($n = 76$) and a chicken farm ($n = 103$) in Guangxi Zhuang Autonomous Region, China. The samples were cultured in LB broth for 12 h and then inoculated onto CHROMagar *Acinetobacter* plates (France) containing tigecycline (2 mg/L) for the selection of isolates. Colonies with different morphologies were selected for PCR screening of tigecycline resistance genes. Species identification was conducted by 16S rRNA gene sequencing. Two tigecycline-resistant strains, BDT2044 and BDT2091, were successfully identified by PCR and sequenced to identify further the tigecycline-resistant genes contained in these strains (see Table S1 in the supplemental material).

**Antimicrobial susceptibility testing.** Antimicrobial susceptibility testing was measured using the broth microdilution method of the Clinical and Laboratory Standards Institute (CLSI); the MICs of ciprofloxacin, imipenem, meropenem, florfenicol, streptomycin, erythromycin, fosfomycin, tetracycline, and tigecycline were determined, and the tigecycline breakpoint was interpreted according to the EUCAST criteria for *Enterobacteriaceae* bacteria (http://www.eucast.org/clinical_breakpoints/). *E. coli* ATCC 25922 was used as the quality control strain.

**Conjugation experiments.** In order to determine the transferability of the resistance genes, we used *Acinetobacter baylyi* ADP1 (rifampicin resistant) as the recipient. Potential transconjugants were selected on LB agar with 2 mg/L tigecycline and 100 mg/L rifampicin. At the same time, we also used *Acinetobacter baumannii* ATCC 19606 (rifampicin resistant) and *E. coli* strains EC600 (rifampicin resistant) and J53 (azide resistant) as the recipient strains to test for conjugation transfer. PCR and 16S rRNA sequencing confirmed the transconjugants. The number of transconjugants obtained per recipient was calculated as the transfer frequency.

**Whole-genome sequencing and bioinformatics analysis.** *A. variabilis* BDT2044, *A. schindleri* BDT2091, and the transconjugants JAT2044 and JAT2091 were subjected to whole-genome sequencing (WGS) using the Nanopore PromethION platform and the Illumina NovaSeq instrument (paired-end [PE], 150-bp format). The Nanopore and Illumina reads were combined to produce data for the genome assembly, which was performed using Unicycler version 0.4.3 (42). The RAST server (http://rast.nmpdr.org) was used to analyze functional genes for annotation and classification. Antibiotic resistance genes were identified using the CGE server ResFinder version 4.1 (http://www.genomicepidemiology.org/services/), and IS genetic elements were identified using ISfinder (https://isfinder.biotoul.fr/). DNAPlotter version 1.11 and Easyfig version 2.2.5 were used to visualize the genetic comparisons (43, 44). A phylogenetic analysis of the amino acid sequences of all Tet(X) proteins was constructed using the neighbor-joining method with MEGA version 11.0.11 (45) using default parameters, and the alignments were performed using ESPript version 3.0 (46).

**Gene cloning and functional analysis.** In order to verify the contribution to tigecycline resistance of different *tet*(X3) variants in bacteria, we amplified the promoter region of *tet*(X3) using PCR technology, with the addition of an EcoRI restriction site at the 5′ end and an XbaI restriction site at the 3′ end (Table S1). Thus, we constructed pBAD24-pro. pBAD24-pro was transformed into *E. coli* DH5α using the heat shock method. Four *tet*(X3) variants were synthesized in BDT2044 and three *tet*(X3) variants in BDT2091, with the addition of a SalI restriction site at the 5′ end and a HindIII restriction site at the 3′ end (Table S1). The recombinant vector pBAD24-pro and other variants were digested with the SalI/HindIII restriction endonucleases and then ligated at 16°C overnight to construct pBAD24-pro-*tet*(X3) variants (New England Biolabs, Ipswich, MA, USA). The recombinant plasmid pBAD24-pro-*tet*(X3) variants

were transformed into *E. coli* JM109 using the heat shock method. In parallel, *tet*(X3) was cloned into pBAD24 to construct pBAD24-*tet*(X3), which was transformed into *E. coli* JM109 using the heat shock method as the positive control.

The transformants were selected on LB agar plates containing 100 mg/L ampicillin; they were verified by PCR and sequencing and then subjected to antimicrobial susceptibility testing (AST) using a broth microdilution method.

**Data availability.** The genome sequence data of *A. variabilis* BDT2044, *A. schindleri* BDT2091, and *A. baylyi* strains JAT2044 and JAT2091 have been submitted to NCBI under the BioSample accession numbers SAMN26686321, SAMN26688062, SAMN30864159, and SAMN30864160, respectively.

## SUPPLEMENTAL MATERIAL

Supplemental material is available online only.
**SUPPLEMENTAL FILE 1**, PDF file, 0.3 MB.

## ACKNOWLEDGMENTS

We warmly thank Jian Sun for sharing the recipient strains *Acinetobacter baylyi* ADP1 (rifampicin resistant), *Acinetobacter baumannii* ATCC 19606 (rifampicin resistant), and *E. coli* EC600 (rifampicin resistant) and J53 (azide resistant). This work was jointly supported by the Key Research and Development Plan of Guangxi, China (AB19245037), Joint Funds of the National Natural Science Foundation of China (U22A20523), and the Major R&D Project of Nanning Qingxiu District (2020005). We have no competing interests to declare.

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
