## [Reviewer comments · Microbiology Spectrum]

Microbiology Spectrum

Identification of novel *tet(X3)* variants resistance to tigecycline in *Acinetobacter* spp.

Yumeng Cheng, Yakun Li, Runhao Yu, Mingxiang Ma, Meng Yang, and Hongbin Si

Corresponding Author(s): Hongbin Si, Guangxi University

Review Timeline:

Submission Date:	April 11, 2022
Editorial Decision:	June 14, 2022
Revision Received:	August 12, 2022
Editorial Decision:	August 19, 2022
Revision Received:	September 15, 2022
Editorial Decision:	September 28, 2022
Revision Received:	October 25, 2022
Accepted:	November 1, 2022

Editor: Katharina Schaufler

Reviewer(s): Disclosure of reviewer identity is with reference to reviewer comments included in decision letter(s). The following individuals involved in review of your submission have agreed to reveal their identity: Wanjiang Zhang (Reviewer #4)

Transaction Report:

DOI: <https://doi.org/10.1128/spectrum.01333-22>

June 14, 2022

Dr. Hongbin Si
Guangxi University
College of Animal Sciences and Technology
Nanning, Guangxi
China

Re: Spectrum01333-22 (Identification of novel *tet(X3)* variants resistance to tigecycline in *Acinetobacter* spp.)

Dear Dr. Hongbin Si:

Link Not Available

Sincerely,

Katharina Schaufler

Journals Department
Reviewer comments:

Reviewer #1 (Comments for the Author):

1. Currently, there are no MIC breakpoints for tigecycline against *Acinetobacter* spp in EUCAST, CLSI100, and FDA standards. EUCAST and FDA Antibacterial Susceptibility Test Interpretive Criteria contains tigecycline breakpoints for Enterobacteriaceae. *Acinetobacter* spp refer to this Criteria (tigecycline breakpoints for Enterobacteriaceae) temporarily. The reference for tigecycline's breakpoint judgment for *Acinetobacter* spp can be described more clearly.
2. According to the drug resistance rate and detection rate reported in the literature, *Acinetobacter baumannii* has received the highest clinical attention among *Acinetobacter* spp. It may be more clinically meaningful to study the novel *tet(X3)* variants and whether these genes can be conjugated to *A. baumannii*. Of course, It cannot be excluded that other species within the *Acinetobacter* genus would be the main prevalent species for future infections.

3. As you described, the novel tet(X3) variants could not conjugate transferred with E. coli EC600 and J53 after three repeats. In the natural state, whether novel tet(X3) variants could conjugate with Enterobacteriaceae or Acinetobacter baumannii may require further study.

Reviewer #3 (Comments for the Author):

The manuscript identified seven tet(X3) variants, designated from tet(X3.3) to tet(X3.9), however their amino acid identities compared to tet(X3) were 58.78% 66.13% 22.71% 23.28% 99.7% 15.4% and 99.7%, respectively. Can the amino acid sequences with 15.4%-58.78% homology be named as variants of tet(X3)? The authors need collect all tetX variants to construct a phylogenetic tree for more accurate and scientific naming? Moreover, the English grammar in the paper is rather bad and should be improved. I will suggest the authors consult a native speaker to help you remove the flaws from the manuscript.

Other comments:

Line 35: clinical should be clinic.

Line 47: The reference for tet(M) variant was not cited here.

Line 48: enzyme should be enzymy

Lines 69-70: "However, they could conjugate transferred with Acinetobacter Bailey ADP1, at a frequency of $(5.3 \pm 0.4) \times 10^{-6}$, $(1.2 \pm 0.8) \times 10^{-7}$ cells per recipient cell" The manuscript indicates that both donors have obtained tigeicycline-resistant transconjugants by conjugation. Interestingly, tet(X3) in the donor BDT2044 is located on a gene island in the chromosome. The authors should at least sequence the transconjugant and examine what exactly causes the transfer of the resistant phenotype in transconjugant JAT2044.

For BDT2091, is the pBDT2091-4 a conjugative plasmid? If it is, the authors should label the conjugation region on this plasmid. If it is not, is there any transfer apparatus in this strain? The authors should described this in the text in detail.

Line 79: The authors should check 13 consecutive tet(X3) copies carefully to confirm whether it is correct using the original sequencing data.

Lines 140-141: "The tet(X) variants of tigeicycline-resistant have already been extended from tet(X3) to tet(X15)" is not updated. In 2021, Rong-min Zhang et al. had reported tet(X47). [doi:10.1128/Spectrum.01164-21].

Line 104: I would suggest delete the blaOXA-276 related part including Fig 5.

Line 127-135: It is confusing and should reconstruct the sentence.

Line 151: The word "brone" should be corrected to "borne".

Line 157-159: It is confusing and should reconstruct the sentence.

Line 173: E.coil should be E.coli

Line 174: The word "brone" looks like a spelling mistake ?

Lines 204-205: "Nanopore and Illumina reads were combined to produce data for genome assembly were performed with Unicycler version 0.4.3" The complete sequence of the strain in the manuscript is the result of assembly by the unicycler software, however its sequence contains multiple tandem repeat structures, due to the specificity of this structure (the assembly is prone to mismatches), whether the authors used other means (such as checking the long reads raw data or PCR products) to verify the accuracy of these structures?

Line 354: The source of the strain should be noted in the Figure legend?

Reviewer #4 (Comments for the Author):

1. The format of some references needs to be modified, such as Ref 39.

Staff Comments:

Preparing Revision Guidelines

Please return the manuscript within 60 days; if you cannot complete the modification within this time period, please contact me. If you do not wish to modify the manuscript and prefer to submit it to another journal, please notify me of your decision immediately so that the manuscript may be formally withdrawn from consideration by Microbiology Spectrum.

The manuscript identified seven *tet(X3)* variants, designated from *tet(X3.3)* to *tet(X3.9)*, however their amino acid identities compared to *tet(X3)* were 58.78% 66.13% 22.71 % 23.28% 99.7% 15.4% and 99.7%, respectively. Can the amino acid sequences with 15.4%-58.78% homology be named as variants of *tet(X3)*? The authors need collect all *tetX* variants to construct a phylogenetic tree for more accurate and scientific naming? Moreover, the English grammar in the paper is rather bad and should be improved. I will suggest the authors consult a native speaker to help you remove the flaws from the manuscript.

Other comments:

Line 35: clinical should be clinic.

Line 47: The reference for *tet(M)* variant was not cited here.

Line 48:enzyme should be enzemy

Lines 69-70: “However, they could conjugate transferred with *Acinetobacter Bailey* ADP1, at a frequency of $(5.3 \pm 0.4) \times 10^{-6}$, $(1.2 \pm 0.8) \times 10^{-7}$ cells per recipient cell” The manuscript indicates that both donors have obtained tigecycline-resistant transconjugants by conjugation. Interestingly, *tet(X3)* in the donor BDT2044 is located on a gene island in the chromosome. The authors should at least sequence the transconjugant and examine what exactly causes the transfer of the resistant phenotype in transconjugant JAT2044.

For BDT2091, is the pBDT2091-4 a conjugative plasmid? If it is, the authors should label the conjugation region on this plasmid. If it is not, is there any transfer apparatus in this strain? The authors should described this in the text in detail.

Line 79:The authors should check 13 consecutive *tet(X3)* copies carefully to confirm whether it is correct using the original sequencing data.

Lines 140-141: “The *tet(X)* variants of tigecycline-resistant have already been extended from *tet(X3)* to *tet(X15)*” is not updated. In 2021, Rong-min Zhang et al. had reported *tet(X47)*. [doi:10.1128/Spectrum.01164-21].

Line 104: I would suggest delete the blaOXA-276 related part including Fig 5.

Line 127-135: It is confusing and should reconstruct the sentence.

Line 151: The word “brone” should be corrected to “borne”.

Line 157-159: It is confusing and should reconstruct the sentence.

Line 173: E.coil should be E.coli

Line 174: The word “brone” looks like a spelling mistake ?

Lines 204-205: “Nanopore and Illumina reads were combined to produce data for genome assembly were performed with Unicycler version 0.4.3” The complete sequence of the strain in the manuscript is the result of assembly by the unicycler software, however its sequence contains multiple tandem repeat structures, due to the specificity of this structure (the assembly is prone to mismatches), whether the authors used other means (such as checking the long reads raw data or PCR products) to verify the accuracy of these structures?

Line 354: The source of the strain should be noted in the Figure legend?

Response to Reviewers

(Modifications in the manuscript are highlighted in yellow)

Questions:

Reviewer 1:

1. Currently, there are no MIC breakpoints for tigecycline against *Acinetobacter* spp in EUCAST, CLSI100, and FDA standards. EUCAST and FDA Antibacterial Susceptibility Test Interpretive Criteria contains tigecycline breakpoints for Enterobacteriaceae. *Acinetobacter* spp refer to this Criteria (tigecycline breakpoints for Enterobacteriaceae) temporarily. The reference for tigecycline's breakpoint judgment for *Acinetobacter* spp can be described more clearly.

2. According to the drug resistance rate and detection rate reported in the literature, *Acinetobacter baumannii* has received the highest clinical attention among *Acinetobacter* spp. It may be more clinically meaningful to study the novel tet(X3) variants and whether these genes can be conjugated to *A. baumannii*. Of course, It cannot be excluded that other species within the *Acinetobacter* genus would be the main prevalent species for future infections.

3. As your described, the novel tet(X3) variants could not conjugate transferred with *E. coli* EC600 and J53 after three repeats. In the natural state, whether novel tet(X3) variants could conjugated with Enterobacteriaceae or *Acinetobacter baumannii* may require further study.

Reviewer 3:

The manuscript identified seven tet(X3) variants, designated from tet(X3.3) to tet(X3.9), however their amino acid identities compared to tet(X3) were 58.78% 66.13% 22.71 % 23.28% 99.7% 15.4% and 99.7%, respectively. Can the amino acid sequences with 15.4%-58.78% homology be named as variants of tet(X3)? The authors need collect all tetX variants to construct a phylogenetic tree for more accurate and scientific naming? Moreover, the English grammar in the paper is rather bad and should be improved. I will suggest the authors consult a native speaker to help you remove the flaws from the manuscript.

Other comments:

Line 35: clinical should be clinic.

Line 47: The reference for tet(M) variant was not cited here.

Line 48: enzyme should be enzymy

Lines 69-70: "However, they could conjugate transferred with Acinetobacter Bailey ADP1, at a frequency of $(5.3 \pm 0.4) \times 10^{-6}$, $(1.2 \pm 0.8) \times 10^{-7}$ cells per recipient cell"

The manuscript indicates that both donors have obtained tigecycline-resistant transconjugants by conjugation. Interestingly, tet(X3) in the donor BDT2044 is located on a gene island in the chromosome. The authors should at least sequence the transconjugant and examine what exactly causes the transfer of the resistant phenotype in transconjugant JAT2044.

For BDT2091, is the pBDT2091-4 a conjugative plasmid? If it is, the authors should label the conjugation region on this plasmid. If it is not, is there any transfer apparatus in this strain? The authors should describe this in the text in detail.

Line 79: The authors should check 13 consecutive tet(X3) copies carefully to confirm whether it is correct using the original sequencing data.

Lines 140-141: "The tet(X) variants of tigecycline-resistant have already been extended from tet(X3) to tet(X15)" is not updated. In 2021, Rong-min Zhang et al. had reported tet(X47). [doi:10.1128/Spectrum.01164-21].

Line 104: I would suggest delete the blaOXA-276 related part including Fig 5.

Line 127-135: It is confusing and should reconstruct the sentence.

Line 151: The word "brone" should be corrected to "borne".

Line 157-159: It is confusing and should reconstruct the sentence.

Line 173: E. coil should be E. coli

Line 174: The word "brone" looks like a spelling mistake?

Lines 204-205: "Nanopore and Illumina reads were combined to produce data for genome assembly were performed with Unicycler version 0.4.3" The complete sequence of the strain in the manuscript is the result of assembly by the unicycler software, however its sequence contains multiple tandem repeat structures, due to the specificity of this structure (the assembly is prone to mismatches), whether the authors used other means (such as checking the long reads raw data or PCR products) to verify the accuracy of these structures?

Line 354: The source of the strain should be noted in the Figure legend?

Reviewer 4:

1. The format of some references needs to be modified, such as Ref 39.

Response to Reviewers

Reviewer 1:

1. Currently, there are no MIC breakpoints for tigecycline against *Acinetobacter* spp in EUCAST, CLSI100, and FDA standards. EUCAST and FDA Antibacterial Susceptibility Test Interpretive Criteria contains tigecycline breakpoints for Enterobacteriaceae. *Acinetobacter* spp refer to this Criteria (tigecycline breakpoints for Enterobacteriaceae) temporarily. The reference for tigecycline's breakpoint judgment for *Acinetobacter* spp can be described more clearly.

As suggested by the reviewer, the content of "Antimicrobial susceptibility testing" has been modified, and the modifications can be seen on lines 199-202.

2. According to the drug resistance rate and detection rate reported in the literature, *Acinetobacter baumannii* has received the highest clinical attention among *Acinetobacter* spp. It may be more clinically meaningful to study the novel tet(X3) variants and whether these genes can be conjugated to *A. baumannii*. Of course, It cannot be excluded that other species within the *Acinetobacter* genus would be the main prevalent species for future infections.

According to the drug resistance rate and detection rate reported in the literature, *Acinetobacter baumannii* does receive the highest clinical attention. Since our research group did not have a suitable *A. baumannii* as the recipient strain, we were unable to investigate whether the tet(X3) variants could be conjugated to *A. baumannii*. Meanwhile, according to the method of "genetic diversity and characteristics of high-level tigecycline resistance Tet(X) in *Acinetobacter* species", the novel tet(X3) variants could be successfully transferred into *A. baylyi* ADP1, which also has some clinically meaningful.

3. As your described, the novel tet(X3) variants could not conjugate transferred with *E. coli* EC600 and J53 after three repeats. In the natural state, whether novel tet(X3) variants could conjugated with Enterobacteriaceae or *Acinetobacter baumannii* may require further study.

In this research, the novel tet(X3) variants could not conjugate transferred with *E. coli* EC600 and

J53 after three repeats. In the natural state, whether novel *tet(X3)* variants could conjugated with *Enterobacteriaceae* or *Acinetobacter baumannii* may require further study.

Reviewer 3:

The manuscript identified seven tet(X3) variants, designated from tet(X3.3) to tet(X3.9), however their amino acid identities compared to tet(X3) were 58.78% 66.13% 22.71 % 23.28% 99.7% 15.4% and 99.7%, respectively. Can the amino acid sequences with 15.4%-58.78% homology be named as variants of tet(X3)? The authors need collect all tetX variants to construct a phylogenetic tree for more accurate and scientific naming?

As suggested by the reviewer, phylogenetic tree for amino acid sequences of all Tet(X)s was constructed using neighbor joining by using Mega XI Version 11.0.11, and the modifications can be seen on lines 77-78, 106-108, 221-223, Figure 5, Ref. 45 and Ref. 46.

Moreover, the English grammar in the paper is rather bad and should be improved. I will suggest the authors consult a native speaker to help you remove the flaws from the manuscript.

As suggested by the reviewer, the use of English language throughout the article has been revised. The “*Acinetobacter Bailey ADP1*” that appears throughout the article has also been changed to “*Acinetobacter Baylyi ADP1*”.

Other comments:

Line 35: clinical should be clinic.

As suggested by the reviewer, the content of “Introduction” has been modified, the “clinical” has been changed to “clinic”, and the modifications can be seen on lines 35.

Line 47: The reference for tet(M) variant was not cited here.

As suggested by the reviewer, the content of “Introduction” has been modified, and the modifications can be seen on lines 46-48 and Ref. 26.

Line 48:enzyme should be enzemy

As suggested by the reviewer, the “enzyme” that appears throughout the article has been changed to “enzymes”, and the modifications can be seen on lines 48.

Lines 69-70: "However, they could conjugate transferred with Acinetobacter Bailey ADP1,

at a frequency of $(5.3 \pm 0.4) \times 10^{-6}$, $(1.2 \pm 0.8) \times 10^{-7}$ cells per recipient cell" The manuscript indicates that both donors have obtained tigeicycline-resistant transconjugants by conjugation. Interestingly, *tet(X3)* in the donor BDT2044 is located on a gene island in the chromosome. The authors should at least sequence the transconjugant and examine what exactly causes the transfer of the resistant phenotype in transconjugant JAT2044.

For BDT2091, is the pBDT2091-4 a conjugative plasmid? If it is, the authors should label the conjugation region on this plasmid. If it is not, is there any transfer apparatus in this strain? The authors should describe this in the text in detail.

As suggested by the reviewer, we sequenced the transconjugants JAT2044 and JAT2091 and found that 5,117-bp circular intermediate (*res-ΔISCR2-ISVsa3-xerD-tet(X3)*) forming a minicircle plasmid in the transconjugants JAT2044 to mediate the resistance of the tigeicycline. And part of the region of pBDT2091-4 plasmid was re-formed a 27,995-bp plasmid and named pJAT2091. The tigeicycline resistance region could be transferred into the recipient strain ADP1 by the mobilization of the conjugative plasmid pBDT2091-6. The modifications can be seen on lines 98-99, 114, 121-132, 168-171 and 212-214.

Line 79: The authors should check 13 consecutive *tet(X3)* copies carefully to confirm whether it is correct using the original sequencing data.

As suggested by the reviewer, we performed confirmation, but due to the large repeat sequences, we could only confirm that up to 7 consecutive *tet(X3)* copies results were correct using the raw sequencing data, and finally confirmed that 13 consecutive *tet(X3)* copies structures were correct by sequence splicing the results.

Lines 140-141: "The *tet(X)* variants of tigeicycline-resistant have already been extended from *tet(X3)* to *tet(X15)*" is not updated. In 2021, Rong-min Zhang et al. had reported *tet(X47)*. [doi:10.1128/Spectrum.01164-21].

As suggested by the reviewer, the content of "Introduction" and "Discussion" have been modified, and the modifications can be seen on Ref. 22 and lines 43-46 and 148-149.

Line 104: I would suggest delete the *blaOXA-276* related part including Fig 5.

As suggested by the reviewer, the content of "the *blaOXA-276* related part including Fig 5" have been deleted.

Line 127-135: It is confusing and should reconstruct the sentence.

As suggested by the reviewer, this sentence has been revised, and the modifications can be seen on lines 140-144.

Line 151: The word "brone" should be corrected to "borne".

As suggested by the reviewer, the “brone” that appears throughout the article has been changed to “borne”, and the modifications can be seen on lines 158-160.

Line 157-159: It is confusing and should reconstruct the sentence.

As suggested by the reviewer, this sentence has been revised, and the modifications can be seen on lines 163-168.

Line 173: E.coil should be E.coli

As suggested by the reviewer, the “E.coil” that appears throughout the article has been changed to “*E.coli*”, and the modifications can be seen on lines 180-182.

Line 174: The word "brone" looks like a spelling mistake ?

As suggested by the reviewer, the “brone” that appears throughout the article has been changed to “borne”, and the modifications can be seen on lines 158-160.

Lines 204-205: "Nanopore and Illumina reads were combined to produce data for genome assembly were performed with Unicycler version 0.4.3" The complete sequence of the strain in the manuscript is the result of assembly by the unicycler software, however its sequence contains multiple tandem repeat structures, due to the specificity of this structure (the assembly is prone to mismatches), whether the authors used other means (such as checking the long reads raw data or PCR products) to verify the accuracy of these structures?

As suggested by the reviewer, the results of genome assembly with Unicycler version 0.4.3 were quality-checked. We also verify the accuracy of these structures by examining the long read raw data, but due to the high similarity between the *tet(X3)* variant sequences, it was inconvenient to verify them by PCR amplification.

Line 354: The source of the strain should be noted in the Figure legend?

As suggested by the reviewer, the content of "Figure legend" has been modified, and the modifications can be seen on lines 386 and 404.

Reviewer 4:

1. The format of some references needs to be modified, such as Ref 39.

As suggested by the reviewer, the content of "references" has been modified, and the format of references has been uniformly modified to use the ASM literature format.

In addition to this, english language, each corresponding figure and reference has been modified according to the revision of the content in the manuscript.

August 19, 2022

Dr. Hongbin Si
Guangxi University
College of Animal Sciences and Technology
Nanning, Guangxi
China

Re: Spectrum01333-22R1 (Identification of novel *tet(X3)* variants resistance to tigecycline in *Acinetobacter* spp.)

Dear Dr. Hongbin Si:

Some of the reviewers' comments have not been adequately addressed. For example, conjugation experiments in a clinical *A. baumannii* strain should be performed indeed. The strain could be commercially purchased or obtained in the context of a collaboration. Please provide careful responses to the reviewers' suggestions! In addition, English grammar (and style) of the manuscript is unacceptable.

Link Not Available

Sincerely,

Katharina Schaufler

Journals Department
Reviewer comments:

Staff Comments:

Preparing Revision Guidelines

To submit your modified manuscript, log onto the eJP submission site at <https://spectrum.msubmit.net/cgi-bin/main.plex>. Go to Author Tasks and click the appropriate manuscript title to begin the revision process. The information that you entered when you

first submitted the paper will be displayed. Please update the information as necessary. Here are a few examples of required updates that authors must address:

Please return the manuscript within 60 days; if you cannot complete the modification within this time period, please contact me. If you do not wish to modify the manuscript and prefer to submit it to another journal, please notify me of your decision immediately so that the manuscript may be formally withdrawn from consideration by Microbiology Spectrum.

Response to Reviewers

(Modifications in the manuscript are highlighted in yellow)

Questions:

Reviewer 1:

1. Currently, there are no MIC breakpoints for tigecycline against *Acinetobacter* spp in EUCAST, CLSI100, and FDA standards. EUCAST and FDA Antibacterial Susceptibility Test Interpretive Criteria contains tigecycline breakpoints for Enterobacteriaceae. *Acinetobacter* spp refer to this Criteria (tigecycline breakpoints for Enterobacteriaceae) temporarily. The reference for tigecycline's breakpoint judgment for *Acinetobacter* spp can be described more clearly.

2. According to the drug resistance rate and detection rate reported in the literature, *Acinetobacter baumannii* has received the highest clinical attention among *Acinetobacter* spp. It may be more clinically meaningful to study the novel tet(X3) variants and whether these genes can be conjugated to *A. baumannii*. Of course, It cannot be excluded that other species within the *Acinetobacter* genus would be the main prevalent species for future infections.

3. As your described, the novel tet(X3) variants could not conjugate transferred with *E. coli* EC600 and J53 after three repeats. In the natural state, whether novel tet(X3) variants could conjugated with Enterobacteriaceae or *Acinetobacter baumannii* may require further study.

Reviewer 3:

The manuscript identified seven tet(X3) variants, designated from tet(X3.3) to tet(X3.9), however their amino acid identities compared to tet(X3) were 58.78% 66.13% 22.71 % 23.28% 99.7% 15.4% and 99.7%, respectively. Can the amino acid sequences with 15.4%-58.78% homology be named as variants of tet(X3)? The authors need collect all tetX variants to construct a phylogenetic tree for more accurate and scientific naming? Moreover, the English grammar in the paper is rather bad and should be improved. I will suggest the authors consult a native speaker to help you remove the flaws from the manuscript.

Other comments:

Line 35: clinical should be clinic.

Line 47: The reference for tet(M) variant was not cited here.

Line 48: enzyme should be enzymy

Lines 69-70: "However, they could conjugate transferred with Acinetobacter Bailey ADP1, at a frequency of $(5.3 \pm 0.4) \times 10^{-6}$, $(1.2 \pm 0.8) \times 10^{-7}$ cells per recipient cell"

The manuscript indicates that both donors have obtained tigecycline-resistant transconjugants by conjugation. Interestingly, tet(X3) in the donor BDT2044 is located on a gene island in the chromosome. The authors should at least sequence the transconjugant and examine what exactly causes the transfer of the resistant phenotype in transconjugant JAT2044.

For BDT2091, is the pBDT2091-4 a conjugative plasmid? If it is, the authors should label the conjugation region on this plasmid. If it is not, is there any transfer apparatus in this strain? The authors should describe this in the text in detail.

Line 79: The authors should check 13 consecutive tet(X3) copies carefully to confirm whether it is correct using the original sequencing data.

Lines 140-141: "The tet(X) variants of tigecycline-resistant have already been extended from tet(X3) to tet(X15)" is not updated. In 2021, Rong-min Zhang et al. had reported tet(X47). [doi:10.1128/Spectrum.01164-21].

Line 104: I would suggest delete the blaOXA-276 related part including Fig 5.

Line 127-135: It is confusing and should reconstruct the sentence.

Line 151: The word "brone" should be corrected to "borne".

Line 157-159: It is confusing and should reconstruct the sentence.

Line 173: E. coil should be E. coli

Line 174: The word "brone" looks like a spelling mistake?

Lines 204-205: "Nanopore and Illumina reads were combined to produce data for genome assembly were performed with Unicycler version 0.4.3" The complete sequence of the strain in the manuscript is the result of assembly by the unicycler software, however its sequence contains multiple tandem repeat structures, due to the specificity of this structure (the assembly is prone to mismatches), whether the authors used other means (such as checking the long reads raw data or PCR products) to verify the accuracy of these structures?

Line 354: The source of the strain should be noted in the Figure legend?

Reviewer 4:

1. The format of some references needs to be modified, such as Ref 39.

Response to Reviewers

Reviewer 1:

1. Currently, there are no MIC breakpoints for tigecycline against *Acinetobacter* spp in EUCAST, CLSI100, and FDA standards. EUCAST and FDA Antibacterial Susceptibility Test Interpretive Criteria contains tigecycline breakpoints for Enterobacteriaceae. *Acinetobacter* spp refer to this Criteria (tigecycline breakpoints for Enterobacteriaceae) temporarily. The reference for tigecycline's breakpoint judgment for *Acinetobacter* spp can be described more clearly.

As suggested by the reviewer, the content of "Antimicrobial susceptibility testing" has been modified, and the modifications can be seen on lines 201-204.

2. According to the drug resistance rate and detection rate reported in the literature, *Acinetobacter baumannii* has received the highest clinical attention among *Acinetobacter* spp. It may be more clinically meaningful to study the novel tet(X3) variants and whether these genes can be conjugated to *A. baumannii*. Of course, It cannot be excluded that other species within the *Acinetobacter* genus would be the main prevalent species for future infections.

According to the drug resistance rate and detection rate reported in the literature, *Acinetobacter baumannii* does receive the highest clinical attention. As suggested by the reviewer to conduct the study, the strains could not be successfully transferred into the recipient *Acinetobacter baumannii*, the modifications can be seen on lines 116-118, 208-211, 245-248. Meanwhile, according to the methodology of the literature "genetic diversity and characteristics of high-level tigecycline resistance Tet(X) in *Acinetobacter* species", the novel tet(X3) variants could be successfully transferred into *A. baylyi* ADP1, which also has some clinically meaningful.

3. As your described, the novel tet(X3) variants could not conjugate transferred with *E. coli* EC600 and J53 after three repeats. In the natural state, whether novel tet(X3) variants could conjugated with Enterobacteriaceae or *Acinetobacter baumannii* may require further study.

In this research, the novel tet(X3) variants could not conjugate transferred with *Acinetobacter baumannii* ATCC19606, *E. coli* EC600 and J53, the modifications can be seen on lines 116-118,

208-211, 245-248.

Reviewer 3:

The manuscript identified seven tet(X3) variants, designated from tet(X3.3) to tet(X3.9), however their amino acid identities compared to tet(X3) were 58.78% 66.13% 22.71 % 23.28% 99.7% 15.4% and 99.7%, respectively. Can the amino acid sequences with 15.4%-58.78% homology be named as variants of tet(X3)? The authors need collect all tetX variants to construct a phylogenetic tree for more accurate and scientific naming?

As suggested by the reviewer, phylogenetic tree for amino acid sequences of all Tet(X)s was constructed using neighbor joining by using Mega XI Version 11.0.11, and the modifications can be seen on lines 108-114, 224-226, Figure 5, Ref. 44 and Ref. 45.

Moreover, the English grammar in the paper is rather bad and should be improved. I will suggest the authors consult a native speaker to help you remove the flaws from the manuscript.

As suggested by the reviewer, the use of English language throughout the article has been revised. The “*Acinetobacter Bailey ADP1*” that appears throughout the article has also been changed to “*Acinetobacter Baylyi ADP1*”.

Other comments:

Line 35: clinical should be clinic.

As suggested by the reviewer, the content of “Introduction” has been modified, the “clinical” has been changed to “clinic”, and the modifications can be seen on lines 36.

Line 47: The reference for tet(M) variant was not cited here.

As suggested by the reviewer, the content of “Introduction” has been modified, and the modifications can be seen on lines 47-48 and Ref. 26.

Line 48:enzyme should be enzemy

As suggested by the reviewer, the “enzyme” that appears throughout the article has been changed to “enzymes”, and the modifications can be seen on lines 49.

Lines 69-70: "However, they could conjugate transferred with *Acinetobacter Bailey ADP1*, at a frequency of $(5.3 \pm 0.4) \times 10^{-6}$, $(1.2 \pm 0.8) \times 10^{-7}$ cells per recipient cell" The manuscript indicates that both donors have obtained tigeicycline-resistant

transconjugants by conjugation. Interestingly, tet(X3) in the donor BDT2044 is located on a gene island in the chromosome. The authors should at least sequence the transconjugant and examine what exactly causes the transfer of the resistant phenotype in transconjugant JAT2044.

For BDT2091, is the pBDT2091-4 a conjugative plasmid? If it is, the authors should label the conjugation region on this plasmid. If it is not, is there any transfer apparatus in this strain? The authors should describe this in the text in detail.

As suggested by the reviewer, we sequenced the transconjugants JAT2044 and JAT2091 and found that 5,117-bp circular intermediate (*res-ΔISCR2-ISVsa3-xerD-tet(X3)*) forming a minicircle plasmid in the transconjugants JAT2044 to mediate the resistance of the tigecycline. And part of the region of pBDT2091-4 plasmid was re-formed a 27,995-bp plasmid and named pJAT2091. The tigecycline resistance region could be transferred into the recipient strain ADP1 by the mobilization of the conjugative plasmid pBDT2091-6. The modifications can be seen on lines 97-98, 115, 122-134, 160-173 and 215-217.

Line 79: The authors should check 13 consecutive tet(X3) copies carefully to confirm whether it is correct using the original sequencing data.

As suggested by the reviewer, we performed confirmation, but due to the large repeat sequences, we could only confirm that up to 7 consecutive *tet(X3)* copies results were correct using the raw sequencing data, and finally confirmed that 13 consecutive *tet(X3)* copies structures were correct by sequence splicing the results.

Lines 140-141: "The tet(X) variants of tigecycline-resistant have already been extended from tet(X3) to tet(X15)" is not updated. In 2021, Rong-min Zhang et al. had reported tet(X47). [doi:10.1128/Spectrum.01164-21].

As suggested by the reviewer, the content of "Introduction" and "Discussion" have been modified, and the modifications can be seen on Ref. 22 and lines 44-47 and 150-153.

Line 104: I would suggest delete the blaOXA-276 related part including Fig 5.

As suggested by the reviewer, the content of "the *blaOXA-276* related part including Fig 5" have been deleted.

Line 127-135: It is confusing and should reconstruct the sentence.

As suggested by the reviewer, this sentence has been revised, and the modifications can be seen on

lines 142-146.

Line 151: The word "brone" should be corrected to "borne".

As suggested by the reviewer, the “brone” that appears throughout the article has been changed to “borne”, and the modifications can be seen on lines 161.

Line 157-159: It is confusing and should reconstruct the sentence.

As suggested by the reviewer, this sentence has been revised, and the modifications can be seen on lines 165-170.

Line 173: E.coil should be E.coli

As suggested by the reviewer, the “E.coil” that appears throughout the article has been changed to “*E.coli*”, and the modifications can be seen on lines 183.

Line 174: The word "brone" looks like a spelling mistake ?

As suggested by the reviewer, the “brone” that appears throughout the article has been changed to “borne”, and the modifications can be seen on lines 161.

Lines 204-205: "Nanopore and Illumina reads were combined to produce data for genome assembly were performed with Unicycler version 0.4.3" The complete sequence of the strain in the manuscript is the result of assembly by the unicycler software, however its sequence contains multiple tandem repeat structures, due to the specificity of this structure (the assembly is prone to mismatches), whether the authors used other means (such as checking the long reads raw data or PCR products) to verify the accuracy of these structures?

As suggested by the reviewer, the results of genome assembly with Unicycler version 0.4.3 were quality-checked. We also verify the accuracy of these structures by examining the long read raw data, but due to the high similarity between the *tet(X3)* variant sequences, it was inconvenient to verify them by PCR amplification.

Line 354: The source of the strain should be noted in the Figure legend?

As suggested by the reviewer, the content of "Figure legend" has been modified, and the modifications can be seen on Figure 1 and Figure 2 legend.

Reviewer 4:

1. The format of some references needs to be modified, such as Ref 39.

As suggested by the reviewer, the content of "references" has been modified, and the format of

references has been uniformly modified to use the ASM literature format.

In addition to this, english language, each corresponding figure and reference has been modified according to the revision of the content in the manuscript.

September 28, 2022

Dr. Hongbin Si
Guangxi University
College of Animal Sciences and Technology
Nanning, Guangxi
China

Re: Spectrum01333-22R2 (Identification of novel *tet(X3)* variants resistance to tigecycline in *Acinetobacter* spp.)

Dear Dr. Hongbin Si:

Quality of English grammar and style is still not sufficient! Please revise accordingly, I would highly recommend a professional editing service. Also, please explain why most of your conjugation experiments did not work. You might even want to consider including this in the discussion section.

Thank you for submitting your manuscript to Microbiology Spectrum. As you will see your paper is very close to acceptance. Please modify the manuscript along the lines I have recommended. As these revisions are quite minor, I expect that you should be able to turn in the revised paper in less than 30 days, if not sooner. If your manuscript was reviewed, you will find the reviewers' comments below.

When submitting the revised version of your paper, please provide (1) point-by-point responses to the issues raised by the reviewers as file type "Response to Reviewers," not in your cover letter, and (2) a PDF file that indicates the changes from the original submission (by highlighting or underlining the changes) as file type "Marked Up Manuscript - For Review Only". Please use this link to submit your revised manuscript. Detailed instructions on submitting your revised paper are below.

Link Not Available

Sincerely,

Katharina Schaufler

Reviewer comments:

Preparing Revision Guidelines

Please return the manuscript within 60 days; if you cannot complete the modification within this time period, please contact me. If you do not wish to modify the manuscript and prefer to submit it to another journal, please notify me of your decision immediately so that the manuscript may be formally withdrawn from consideration by Microbiology Spectrum.

Response to Reviewers

(Modifications in the manuscript are highlighted in yellow)

Questions:

Reviewer 1:

1. Currently, there are no MIC breakpoints for tigecycline against *Acinetobacter* spp in EUCAST, CLSI100, and FDA standards. EUCAST and FDA Antibacterial Susceptibility Test Interpretive Criteria contains tigecycline breakpoints for Enterobacteriaceae. *Acinetobacter* spp refer to this Criteria (tigecycline breakpoints for Enterobacteriaceae) temporarily. The reference for tigecycline's breakpoint judgment for *Acinetobacter* spp can be described more clearly.

2. According to the drug resistance rate and detection rate reported in the literature, *Acinetobacter baumannii* has received the highest clinical attention among *Acinetobacter* spp. It may be more clinically meaningful to study the novel tet(X3) variants and whether these genes can be conjugated to *A. baumannii*. Of course, It cannot be excluded that other species within the *Acinetobacter* genus would be the main prevalent species for future infections.

3. As your described, the novel tet(X3) variants could not conjugate transferred with *E. coli* EC600 and J53 after three repeats. In the natural state, whether novel tet(X3) variants could conjugated with Enterobacteriaceae or *Acinetobacter baumannii* may require further study.

Reviewer 3:

The manuscript identified seven tet(X3) variants, designated from tet(X3.3) to tet(X3.9), however their amino acid identities compared to tet(X3) were 58.78% 66.13% 22.71 % 23.28% 99.7% 15.4% and 99.7%, respectively. Can the amino acid sequences with 15.4%-58.78% homology be named as variants of tet(X3)? The authors need collect all tetX variants to construct a phylogenetic tree for more accurate and scientific naming? Moreover, the English grammar in the paper is rather bad and should be improved. I will suggest the authors consult a native speaker to help you remove the flaws from the manuscript.

Other comments:

Line 35: clinical should be clinic.

Line 47: The reference for tet(M) variant was not cited here.

Line 48: enzyme should be enzymy

Lines 69-70: "However, they could conjugate transferred with Acinetobacter Bailey ADP1, at a frequency of $(5.3 \pm 0.4) \times 10^{-6}$, $(1.2 \pm 0.8) \times 10^{-7}$ cells per recipient cell"

The manuscript indicates that both donors have obtained tigecycline-resistant transconjugants by conjugation. Interestingly, tet(X3) in the donor BDT2044 is located on a gene island in the chromosome. The authors should at least sequence the transconjugant and examine what exactly causes the transfer of the resistant phenotype in transconjugant JAT2044.

For BDT2091, is the pBDT2091-4 a conjugative plasmid? If it is, the authors should label the conjugation region on this plasmid. If it is not, is there any transfer apparatus in this strain? The authors should describe this in the text in detail.

Line 79: The authors should check 13 consecutive tet(X3) copies carefully to confirm whether it is correct using the original sequencing data.

Lines 140-141: "The tet(X) variants of tigecycline-resistant have already been extended from tet(X3) to tet(X15)" is not updated. In 2021, Rong-min Zhang et al. had reported tet(X47). [doi:10.1128/Spectrum.01164-21].

Line 104: I would suggest delete the blaOXA-276 related part including Fig 5.

Line 127-135: It is confusing and should reconstruct the sentence.

Line 151: The word "brone" should be corrected to "borne".

Line 157-159: It is confusing and should reconstruct the sentence.

Line 173: E. coil should be E. coli

Line 174: The word "brone" looks like a spelling mistake?

Lines 204-205: "Nanopore and Illumina reads were combined to produce data for genome assembly were performed with Unicycler version 0.4.3" The complete sequence of the strain in the manuscript is the result of assembly by the unicycler software, however its sequence contains multiple tandem repeat structures, due to the specificity of this structure (the assembly is prone to mismatches), whether the authors used other means (such as checking the long reads raw data or PCR products) to verify the accuracy of these structures?

Line 354: The source of the strain should be noted in the Figure legend?

Reviewer 4:

1. The format of some references needs to be modified, such as Ref 39.

Editor reviews:

Please explain why most of your conjugation experiments did not work. You might even want to consider including this in the discussion section.

Response to Reviewers

Reviewer 1:

1. Currently, there are no MIC breakpoints for tigecycline against *Acinetobacter* spp in EUCAST, CLSI100, and FDA standards. EUCAST and FDA Antibacterial Susceptibility Test Interpretive Criteria contains tigecycline breakpoints for Enterobacteriaceae. *Acinetobacter* spp refer to this Criteria (tigecycline breakpoints for Enterobacteriaceae) temporarily. The reference for tigecycline's breakpoint judgment for *Acinetobacter* spp can be described more clearly.

As suggested by the reviewer, the content of "Antimicrobial susceptibility testing" has been modified, and the modifications can be seen on lines **212-215**.

2. According to the drug resistance rate and detection rate reported in the literature, *Acinetobacter baumannii* has received the highest clinical attention among *Acinetobacter* spp. It may be more clinically meaningful to study the novel tet(X3) variants and whether these genes can be conjugated to *A. baumannii*. Of course, It cannot be excluded that other species within the *Acinetobacter* genus would be the main prevalent species for future infections.

According to the drug resistance rate and detection rate reported in the literature, *Acinetobacter baumannii* does receive the highest clinical attention. As suggested by the reviewer to conduct the study, the strains could not be successfully transferred into the recipient *Acinetobacter baumannii*, the modifications can be seen on lines **118-120, 219-222, 262-264**. Meanwhile, according to the methodology of the literature "Genetic diversity and characteristics of high-level tigecycline resistance Tet(X) in *Acinetobacter* species", the novel tet(X3) variants could be successfully transferred into *A. baylyi* ADP1, which also has some clinically meaningful.

3. As your described, the novel tet(X3) variants could not conjugate transferred with *E. coli*

EC600 and J53 after three repeats. In the natural state, whether novel tet(X3) variants could conjugated with Enterobacteriaceae or acinetobacter baumannii may require further study.

In this research, the novel *tet(X3)* variants could not conjugate transferred with *Acinetobacter baumannii* ATCC19606, *E. coli* EC600 and J53, the modifications can be seen on lines 118-120, 219-222, 262-264.

Reviewer 3:

The manuscript identified seven tet(X3) variants, designated from tet(X3.3) to tet(X3.9), however their amino acid identities compared to tet(X3) were 58.78% 66.13% 22.71 % 23.28% 99.7% 15.4% and 99.7%, respectively. Can the amino acid sequences with 15.4%-58.78% homology be named as variants of tet(X3)? The authors need collect all tetX variants to construct a phylogenetic tree for more accurate and scientific naming?

As suggested by the reviewer, phylogenetic tree for amino acid sequences of all Tet(X)s was constructed using neighbor joining by using Mega XI Version 11.0.11, and the modifications can be seen on lines 110-116, 235-236, Figure 5, Ref. 45 and Ref. 46.

Moreover, the English grammar in the paper is rather bad and should be improved. I will suggest the authors consult a native speaker to help you remove the flaws from the manuscript.

As suggested by the reviewer, the use of English language throughout the article has been revised. The “*Acinetobacter Bailey ADP1*” that appears throughout the article has also been changed to “*Acinetobacter Baylyi ADP1*”.

Other comments:

Line 35: clinical should be clinic.

As suggested by the reviewer, the content of “Introduction” has been modified, the “clinical” has been changed to “clinic”, and the modifications can be seen on lines 36.

Line 47: The reference for tet(M) variant was not cited here.

As suggested by the reviewer, the content of “Introduction” has been modified, and the modifications can be seen on lines 47-48 and Ref. 26.

Line 48:enzyme should be enzemy

As suggested by the reviewer, the “enzyme” that appears throughout the article has been changed

to “enzymes”, and the modifications can be seen on lines 49.

Lines 69-70: "However, they could conjugate transferred with Acinetobacter Bailey ADP1, at a frequency of $(5.3 \pm 0.4) \times 10^{-6}$, $(1.2 \pm 0.8) \times 10^{-7}$ cells per recipient cell" The manuscript indicates that both donors have obtained tigecycline-resistant transconjugants by conjugation. Interestingly, tet(X3) in the donor BDT2044 is located on a gene island in the chromosome. The authors should at least sequence the transconjugant and examine what exactly causes the transfer of the resistant phenotype in transconjugant JAT2044.

For BDT2091, is the pBDT2091-4 a conjugative plasmid? If it is, the authors should label the conjugation region on this plasmid. If it is not, is there any transfer apparatus in this strain? The authors should describe this in the text in detail.

As suggested by the reviewer, we sequenced the transconjugants JAT2044 and JAT2091 and found that 5,117-bp circular intermediate (*res-ΔISCR2-ISVsa3-xerD-tet(X3)*) forming a minicircle plasmid in the transconjugants JAT2044 to mediate the resistance of the tigecycline. And part of the region of pBDT2091-4 plasmid was re-formed a 27,995-bp plasmid and named pJAT2091. The tigecycline resistance region could be transferred into the recipient strain ADP1 by the mobilization of the conjugative plasmid pBDT2091-6. The modifications can be seen on lines 98-99, 117, 124-136, 173-181 and 226-228.

Line 79: The authors should check 13 consecutive tet(X3) copies carefully to confirm whether it is correct using the original sequencing data.

As suggested by the reviewer, we performed confirmation, but due to the large repeat sequences, we could only confirm that up to 7 consecutive *tet(X3)* copies results were correct using the raw sequencing data, and finally confirmed that 13 consecutive *tet(X3)* copies structures were correct by sequence splicing the results.

Lines 140-141: "The tet(X) variants of tigecycline-resistant have already been extended from tet(X3) to tet(X15)" is not updated. In 2021, Rong-min Zhang et al. had reported tet(X47). [doi:10.1128/Spectrum.01164-21].

As suggested by the reviewer, the content of “Introduction” and “Discussion” have been modified, and the modifications can be seen on Ref. 22 and lines 44-47 and 152-153.

Line 104: I would suggest delete the blaOXA-276 related part including Fig 5.

As suggested by the reviewer, the content of "the *bla*OXA-276 related part including Fig 5" have been deleted.

Line 127-135: It is confusing and should reconstruct the sentence.

As suggested by the reviewer, this sentence has been revised, and the modifications can be seen on lines 144-148.

Line 151: The word "brone" should be corrected to "borne".

As suggested by the reviewer, the "brone" that appears throughout the article has been changed to "borne", and the modifications can be seen on lines 163.

Line 157-159: It is confusing and should reconstruct the sentence.

As suggested by the reviewer, this sentence has been revised, and the modifications can be seen on lines 176-178.

Line 173: E.coil should be E.coli

As suggested by the reviewer, the "E.coil" that appears throughout the article has been changed to "*E.coli*", and the modifications can be seen on lines 192.

Line 174: The word "brone" looks like a spelling mistake ?

As suggested by the reviewer, the "brone" that appears throughout the article has been changed to "borne", and the modifications can be seen on lines 163.

Lines 204-205: "Nanopore and Illumina reads were combined to produce data for genome assembly were performed with Unicycler version 0.4.3" The complete sequence of the strain in the manuscript is the result of assembly by the unicycler software, however its sequence contains multiple tandem repeat structures, due to the specificity of this structure (the assembly is prone to mismatches), whether the authors used other means (such as checking the long reads raw data or PCR products) to verify the accuracy of these structures?

As suggested by the reviewer, the results of genome assembly with Unicycler version 0.4.3 were quality-checked. We also verify the accuracy of these structures by examining the long read raw data, but due to the high similarity between the *tet*(X3) variant sequences, it was inconvenient to verify them by PCR amplification.

Line 354: The source of the strain should be noted in the Figure legend?

As suggested by the reviewer, the content of "Figure legend" has been modified, and the modifications can be seen on lines 402-403 and 412-414.

Reviewer 4:

1. The format of some references needs to be modified, such as Ref 39.

As suggested by the reviewer, the content of "references" has been modified, and the format of references has been uniformly modified to use the ASM literature format.

Editor reviews:

Please explain why most of your conjugation experiments did not work. You might even want to consider including this in the discussion section.

As suggested by the editor, the content of "discussion" has been modified, and the modifications can be seen on lines **168-173**.

In addition to this, English grammar and style, each corresponding figure and references has been modified according to the revision of the content in the manuscript.

November 1, 2022

Dr. Hongbin Si
Guangxi University
College of Animal Sciences and Technology
Nanning, Guangxi
China

Re: Spectrum01333-22R3 (Identification of novel *tet(X3)* variants resistance to tigecycline in *Acinetobacter* spp.)

Dear Dr. Hongbin Si:

Your manuscript has been accepted, and I am forwarding it to the ASM Journals Department for publication. You will be notified when your proofs are ready to be viewed.

Sincerely,

Katharina Schaufler
Editor, Microbiology Spectrum
